# A Simple Loss Function for Convergent Algorithm Synthesis using RNNs

**Alexandre Salle**
VTEX
Porto Alegre, BR
`alexandre.salle@vtex.com`

**Shervin Malmasi**
Amazon Inc
Seattle, USA
`malmasi@amazon.com`

## Abstract

Running a Recurrent Neural Network (RNN) over the same input multiple times, or *iterative reasoning*, enables logical extrapolation, where a model can be run on problems larger than the models were trained on. The loss function used to train these networks has a profound impact on their extrapolation ability. In this paper, we propose using a simple loss function called the Delta Loss (Salle & Prates, 2019). We show[1] that the Delta Loss, like the state-of-the-art Progressive Loss (Bansal et al., 2022), leads to convergent algorithm synthesis, but with a simpler formulation, increased training efficiency,[2] and greater robustness.

## 1 Introduction

Iterative reasoning is a fundamental human ability that we use every day to solve problems. For example, when trying to solve a puzzle, a person would probably start by trying different pieces and see if they fit together. If they don't fit, they would put them back and try different ones. This process of trying different pieces and evaluating the results is iterative reasoning.

Recently, Schwarzschild et al. (2021) showed that iterative reasoning can also be applied to artificial neural networks: their Deep Thinking Networks – a form of Recurrent Neural Network (RNN) – can be used to solve prefix-sums, mazes, and chess problems. Subsequent work by Bansal et al. (2022) showed that inputting the original problem at each reasoning step (running the RNN over the same input and computing a new state and output solution) *and* using a different loss function, called the Progressive Loss, are key to achieve extreme extrapolation, solving problem instances much larger/harder than seen during training. Their work is closely related to the ideas in Salle & Prates (2019), who present Think Again Networks and the Delta Loss. Given the similarity between Deep Thinking Networks and Think Again Networks, we posit that the Delta Loss can be used with Deep Thinking Networks and test this idea in this paper. We show that using the Delta Loss enables extreme extrapolation, while being more robust and efficient to train[2] than the Progressive Loss.

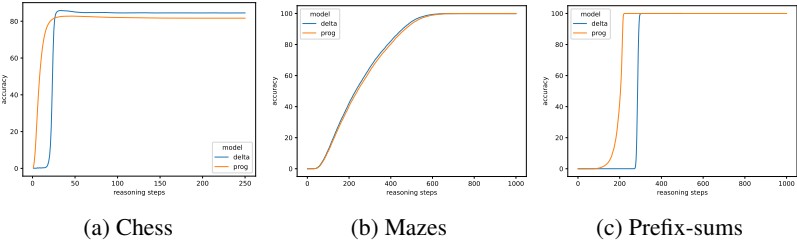

(a) Chess     (b) Mazes     (c) Prefix-sums

Figure 1: Test performance of models trained with the Delta Loss (delta) and Progressive Loss (prog). The models were trained using 30 reasoning steps on 32-bit strings, 9x9 mazes, and easy chess puzzles. The models are tested on hard chess puzzles, 59x59 mazes, and 512-bit strings, *using up to 10x more reasoning steps than used during training*.

---

[1]Our code and models are available at `https://github.com/alexandres/delta-loss`
[2]Except when the Progressive Loss hyper-parameter $\alpha$ is equal to 0 or 1.

## 2    LOSS FUNCTIONS

Given (i.) A state dependent function $F(x, s_t)$ such as an RNN, where $x$ is the input corresponding to a problem instance, and $s_t$ is the state after $t$ reasoning steps (ii.) An arbitrary loss function $\mathcal{L}$ computed over each $1, \ldots, T$ reasoning steps used during training ($T$ set to 30 in all experiments): the Delta Loss ($\mathcal{L}_\Delta$) Salle & Prates (2019), which maximizes the drop in $\mathcal{L}$ between consecutive steps and minimizes the maximum loss over all steps, is defined as:

$$
\begin{aligned}
\mathcal{L}_\Delta(\mathcal{L}, F, T) &= \sum_{1 \leq t \leq T-1} \left( \mathcal{L}(F(x, s_{t+1})) - \mathcal{L}(F(x, s_t)) \right) + \max_{1 \leq t \leq T} \mathcal{L}(F(x, s_t)) \\
&= \mathcal{L}(F(x, s_T)) - \mathcal{L}(F(x, s_1)) + \max_{1 \leq t \leq T} \mathcal{L}(F(x, s_t))
\end{aligned}
\tag{1}
$$

The Progressive Loss ($\mathcal{L}_p$) Bansal et al. (2022) is described in algorithm 1. In contrast to the Delta Loss which *has no hyperparameters*, the Progressive Loss has the hyperparameter $\alpha$. Furthermore, computing the Progressive Loss when $0 < \alpha < 1$ is less efficient than the Delta Loss

---

**Algorithm 1** Progressive Loss

Choose $n \sim U\{0, T-1\}$ and $k \sim U\{1, T-n\}$
Compute $F(x, s_n)$ w/o tracking gradients
Compute $F(x, s_{n+k})$ (additional $k$ steps)
Compute $F(x, s_T)$ with new forward pass of $T$ steps
Compute $\mathcal{L}_p = (1 - \alpha) \cdot \mathcal{L}(F(x, s_T)) + \alpha \cdot \mathcal{L}(F(x, s_{n+k}))$

---

since it requires 2 forward passes: in expectation, this uses 75% more GPU time and memory for computing and storing activations (i.e. in *Chess*, on our GTX1070, the Delta Loss enables batches 78% larger (255 vs. 143), and if using equal batch sizes of 143, computing each batch is 45% faster).

## 3    MATERIALS

We use the exact same datasets, architectures, and code as Bansal et al. (2022). The three datasets used are: (*Chess*) Given a 8x8 chessboard, find the optimal move for a given position. Models are trained on easy chess puzzles and evaluated on hard chess puzzles. (*Mazes*) Given a 2D square image representing a maze, find the shortest path from start to end markers. Models are trained on 9x9 mazes and evaluated on 59x59 mazes. (*Prefix-sums*) The task is to compute prefix-sums of binary strings, where the $j^{th}$ bit of a prefix-sum is the sum of all bits $i \leq j$ in the string, modulo 2. Models are trained on 32-bit strings and evaluated on 512-bit strings. Problems are represented as images, and the same convolutional neural network (CNN) architecture is used for all three datasets, changing only its hyperparameters and dimensionality of the filters (1D for *Prefix-sums*, 2D for *Mazes* and *Chess*). To make the CNN into a state-dependent $F(x, s_t)$, the output from one of the final layers is used as $s_t$, and combined with the input $x$ using a convolutional layer to produce the next input to the network. Both the Delta Loss and Progressive Loss are used with the same network hyperparameters. The architecture and hyperparameters are given in the Appendix.

## 4    RESULTS AND CONCLUSION

Results are shown in fig. 1. On both *Prefix-sums* and *Mazes*, both losses achieve accuracy $\geq 99.9\%$. On the more challenging *Chess* dataset, the Delta Loss significantly outperforms the Progressive Loss (McNemar's test, $p < 0.05$). Both losses exhibit convergence on all datasets. As the number of reasoning steps increase, accuracy stabilizes. Bansal et al. (2022) refer to this convergence as avoiding *overthinking*: unlike the losses tested here, they show that using only the loss from the final reasoning step leads to a collapse in accuracy. Although in our results both the Progressive and Delta Loss perform similarly, results from Bansal et al. (2022) show that the Progressive Loss is highly sensitive to $\alpha$. To circumvent this, they perform a grid-search over $\alpha$ *for each dataset*. Here we report their best configuration for each dataset ($\alpha$=.5,.01,1 on *Chess*, *Mazes*, and *Prefix-sums* respectively). In contrast, the Delta Loss has no hyperparameter and thus no need to tune it for each dataset. We refer to this invariance in hyperparameters as *robustness*. One downside of the Delta Loss is the larger number of steps to convergence on *Chess* and *Prefix-sums* (figs. 1a and 1c). See appendix A.1 for further discussion on this and other future directions.

In sum, we showed that the Delta Loss, like the Progressive Loss, leads to convergent algorithm synthesis, but with a simpler formulation, increased training efficiency,[2] and greater robustness.

URM STATEMENT

The authors acknowledge that at least one key author of this work meets the URM criteria of ICLR 2023 Tiny Papers Track.

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

# A  APPENDIX

## A.1  FUTURE WORK

On the *Chess* and *Prefix-sums* tasks figs. 1a and 1c, the Delta Loss takes longer to converge than the Progressive Loss. In initial experiments on *Prefix-sums*, we observe that during training either reducing the number of reasoning steps $T$ or for each batch drawing $T_{batch} \sim U\{2, T\}$ both significantly reduce the number of steps to convergence when testing. We plan to investigate this and other factors that may contribute to delayed convergence in future work. We also plan to explore other ideas from Salle & Prates (2019) such as (a) allowing periodic divergence during training (taking the *max* operation in eq. (1) every $\lambda$ steps rather than every step), and (b) rather than using only the previous state in the next reasoning step, using a *mixing* function to attend to past states.

## A.2  HARDWARE

All experiments were performed on a desktop with an Intel i5-4430, 32 GB of RAM, and a single Nvidia GTX1070.

## A.3  ARCHITECTURE AND HYPERPARAMETERS

The architectures we use in our experiments are identical to those described in Bansal et al. (2022). We provide a summary here for convenience:

1. Problem instances are represented as binary images $x$, in (*Chess*): $width = 8, height = 8, channels = 12$ (6 channels for each player's piece classes); (*Mazes*, 9x9 or 59x59): $width = \{9 \cdot 2 + 3, 59 \cdot 2 + 3\}, height = \{9 \cdot 2 + 3, 59 \cdot 2 + 3\}, channels = 3$ in (one channel for walls, one channel for start position, one channel for end position, each position is represented by 2x2 pixels and the image has 3 pixel borders); (*Prefix-sums*, 32 or 512 bits): $width = \{32, 512\}, channels = 1$ (single channel for bit values).

2. All convolutional layers use $3 \times 3$ filters (or filters of length three in the 1D case) with stride equal to one and 1-padding, preserving the input width (and height for 2D problems).

3. Initial state $s_1$ is created by a convolutional layer that maps the number of channels in $x$ to the number of channels used in the internal block (item 5).

4. The input to the internal block is formed by a convolutional layer that maps the concatenation of the input and state $[x, s_t]$ channels to the number of channels used in the internal block, followed by a ReLU.

5. The internal block is a standard residual block with four convolutional layers each having the same # of output channels (described in table 1), each followed by a ReLU, and skip connections every two layers .

6. The output of the internal block forms the state $s_{t+1}$. This has the same width (and height in the 2D case) as the input $x$, but has the # of channels specified in table 1. This is then input to item 4 to perform the next reasoning step, turning this entire CNN into an RNN.

7. Next comes the head block which is composed of three convolutional layers with decreasing # of output channels (sizes are in table 1), and ReLUs after the first two layers.

8. The final convolutional layer in the head block has two-channel outputs for binary pixel classification, used to compute the binary cross-entropy loss (move start/end positions are 1s and rest are 0s in *Chess*, shortest path is 1s and rest is 0s in *Mazes*, and targets are 0s and 1s in *Prefix-sums*).

Table 1: Model hyperparameters. The exact same hyperparameters were used for both the Delta Loss and the Progressive Loss.

| Dataset | # output channels in internal block | # output channels in head layers |
|---|---|---|
| Chess | 512 | 32, 8, 2 |
| Mazes | 128 | 32, 8, 2 |
| Prefix-sums | 400 | 400, 200, 2 |

## A.4 DATASET AND TRAINING DETAILS

Dataset sizes are given in table 2.

Table 2: Dataset sizes.

| Dataset | Train | Valid | Test |
|---|---|---|---|
| Chess easy | 480,000 | 120,000 | - |
| Chess hard | - | - | 100,000 |
| Mazes 9x9 | 40,000 | 20,000 | - |
| Mazes 59x59 | - | - | 10,000 |
| Prefix-sums 32 bits | 8,000 | 2,000 | - |
| Prefix-sums 512 bits | - | - | 10,000 |

Training details are given in tables 3 and 4.

Table 3: Training hyperparameters. Dashes indicate that we did not utilize those options. If *LR Throttle* is "Yes", the learning rate of parameters in the internal block is divided by $T$ (reasoning steps used during training).

| Loss | Task | Optimizer | Learning Rate | Decay Schedule | Decay Factor | Warm-Up | Epochs | Clip | LR Throttle |
|---|---|---|---|---|---|---|---|---|---|
| | Chess | Adam | 0.001 | $[100, 110]$ | 0.01 | 3 | 120 | - | Yes |
| Delta | Mazes | Adam | 0.001 | - | - | 10 | 50 | 10.0 | Yes |
| | Prefix Sums | Adam | 0.001 | $[60, 100]$ | 0.01 | 10 | 150 | 0.1 | No |
| | Chess | SGD | 0.010 | $[100, 110]$ | 0.01 | 3 | 120 | - | No |
| Prog | Mazes | Adam | 0.001 | - | - | 10 | 50 | - | Yes |
| | Prefix Sums | Adam | 0.001 | $[60, 100]$ | 0.01 | 10 | 150 | 1.0 | No |

## A.5 DETAILED RESULTS

Detailed results for *Chess*, *Mazes*, and *Prefix-sums* are shown in tables 5 to 7 respectively.

Table 4: Training results. Time per epoch on the machine described in appendix A.2.

| Loss | Task | Final training accuracy (%) | Max validation accuracy (%) | Epoch with max validation (%) | Time per epoch (hours) | Time to epoch with max validation (hours) |
|---|---|---|---|---|---|---|
| | Chess | 100.0 | 95.8 | 130 | 1.57 | 204.5 |
| Delta | Mazes | 100.0 | 100.0 | 35 | 0.15 | 5.15 |
| | Prefix Sums | 100.0 | 100.0 | 6 | 0.01 | 0.07 |
| | Chess | 99.7 | 94.6 | 106 | 2.13 | 226.1 |
| Prog | Mazes | 93.7 | 100.0 | 43 | 0.18 | 7.6 |
| | Prefix Sums | 100.0 | 100.0 | 6 | 0.007 | 0.044 |

Table 5: Steps until maximum accuracy for *Chess*. * indicates the models plotted in fig. 1.

| Loss | $\alpha$ | Steps until max accuracy | Max accuracy (%) |
|---|---|---|---|
| Delta* | - | 33 | 85.83 |
| Progressive | 0.0 | 29 | 82.12 |
| Progressive* | 0.5 | 46 | 82.79 |

Table 6: Steps until maximum accuracy for *Mazes*. * indicates the models plotted in fig. 1.

| Loss | $\alpha$ | Steps until max accuracy | Max accuracy (%) |
|---|---|---|---|
| Delta* | - | 657 | 99.90 |
| Progressive | 0.00 | 999 | 82.72 |
| Progressive* | 0.01 | 808 | 100.0 |

Table 7: Steps until maximum accuracy for *Prefix-sums*. * indicates the models plotted in fig. 1.

| Loss | $\alpha$ | Steps until max accuracy | Max accuracy (%) |
|---|---|---|---|
| Delta* | - | 318 | 100.0 |
| Progressive | 0.0 | 466 | 96.19 |
| Progressive* | 1.0 | 222 | 100.0 |

