# OpenReview forum: "A Simple Loss Function for Convergent Algorithm Synthesis using RNNs"
_ICLR.cc/2023/TinyPapers — Submitted to Tiny Papers @ ICLR 2023_

### Official Review · Reviewer_oASR · 2023-03-27

**Confidence:** 5

**Summary Of Contributions:**

This paper proposes to use the Delta Loss to train RNN to solve iterative reasoning problems. The results show encouraging results of Delta Loss compared to the state-of-the-art Progressive Loss.

**Rating:**

Clear, Correct, and Reproducible (CCR): a submission which meets the reviewing criteria

**Strengths And Weaknesses:**

Strengths
- Empowering (deep) neural networks with the iterative reasoning capability is an important research topic.
- The empirical results are encouraging, showing that Delta Loss can perform comparably with the state-of-the-art progressive loss, while being simpler and more computationally efficient.

Weaknesses
- In Think Again Networks, the authors proposed an abstraction framework with the Delta Loss, and also mentioned RNN as one possible implementation (in the abstract). Given this, the technical contribution and extendibility of this work is quite limited.
- The empirical results are not really strong: on all three tasks, both Delta and Progressive losses perform mostly the same, albeit Delta Loss even took more reasoning steps to converge.
- The authors might consider plotting the accuracy against the training time to highlight the training efficiency of Delta Loss.
- A more detailed discussion on the future work is preferred.

**Suggested Changes:**

Please see Weaknesses.

---

### Official Review · Reviewer_Da5L · 2023-04-01

**Confidence:** 4

**Summary Of Contributions:**

In this submission, the authors run experiments on an alternative training loss (Salle & Prates (2019)) when running a Recurrent Neural Network (RNN) over the same input multiple times (Bansal et al., 2022). They show that the Delta loss enables the same “extreme extrapolation” (scaling to larger instances than seen during training), while being more efficient.

**Rating:**

High Potential (HP): a submission which meets the reviewing criteria and has potential to make an impact on the field

**Strengths And Weaknesses:**

Strength:
- Interesting experiments, a good indication that the usage of the Delta loss can save many resources
- The delta loss performs as well as the proposed loss of Bansal et al.
- The delta loss is hyper-parameter free
- Implementation available

Weaknesses:
- The “robustness” claim is not well-backed
- Limited experimental results only indicate that the Delta loss will outperform

**Suggested Changes:**

I have no particular suggested changes for this submission in this track. I would love to see a deeper analysis on the robustness claim in the future.

---

### Comment · Area_Chair_BNUM · 2023-06-02
**Formatted for archival**

This work meets the threshold for archival, contents the URM statement and is deanonymized.

Congratulations, and good work!

AC BNUM

---

### Meta-Review · Area_Chair_BNUM · 2023-04-06

**Recommendation:** Invite to present
**Confidence:** 4

**Metareview:**

In this work, the authors propose a new loss function for training iterative reasoning networks.  This builds on recent work in this field.  The paper is very well written.

One reviewer questioned the originality of the work.  However actually implementing something proposed abstractly elsewhere is a valuable endeavour.  Seeing a faster method that removes a hyperparameter is likely to engender more follow-up works.

I do echo the sentiment that a more thorough discussion of future work and the weaknesses of this method (or Think Again Networks more generally) would make this paper more usable/interesting/general/extensible.

Overall I think this is a valuable paper to include in the workshop.

**Summary:**

A new loss function for iterative reasoning is proposed.  The loss function matches the performance of existing SotA methods, while bringing additional some benefits.

**Comments And Feedback To The Authors:**

Please take on board the feedback from the reviewers and in the meta review.  The figures can also be made more compact (and with a legible font please!) to clear space for some more discussion.

**Reason For Not Giving A Higher Recommendation:**

Experimental evaluation is a little thin, and as such the contribution is not enormous.

**Reason For Not Giving A Lower Recommendation:**

I agree with the reviewers scores.

---

### Decision · Program_Chairs · 2023-04-08

Invite to present